# Transgenic Expression of dsRNA Targeting the *Pentalonia nigronervosa acetylcholinesterase* Gene in Banana and Plantain Reduces Aphid Populations

**DOI:** 10.3390/plants10040613

**Published:** 2021-03-24

**Authors:** Temitope Jekayinoluwa, Jaindra Nath Tripathi, Benjamin Dugdale, George Obiero, Edward Muge, James Dale, Leena Tripathi

**Affiliations:** 1International Institute of Tropical Agriculture, Biosciences, P.O. Box 30709, Nairobi 00100, Kenya; temitopejekayinoluwa@gmail.com (T.J.); J.Tripathi@cgiar.org (J.N.T.); 2Center for Biotechnology and Bioinformatics, University of Nairobi, 30197, Nairobi 00100, Kenya; george.obiero@uonbi.ac.ke; 3Center for Agriculture and the Bioeconomy, Queensland University of Technology, Brisbane, QLD 4000, Australia; b.dugdale@qut.edu.au (B.D.); j.dale@qut.edu.au (J.D.); 4Department of Biochemistry, University of Nairobi, 30197, Nairobi 00100, Kenya; mugeek@uonbi.ac.ke

**Keywords:** RNA interference, acetylcholinesterase, banana, plantain, banana aphid, sugars, artificial diet

## Abstract

The banana aphid, *Pentalonia nigronervosa*, is the sole insect vector of banana bunchy top virus (BBTV), the causal agent of banana bunchy top disease. The aphid acquires and transmits BBTV while feeding on infected banana plants. RNA interference (RNAi) enables the generation of pest and disease-resistant crops; however, its effectiveness relies on the identification of pivotal gene sequences to target and silence. Acetylcholinesterase (AChE) is an essential enzyme responsible for the hydrolytic metabolism of the neurotransmitter acetylcholine in animals. In this study, the *AChE* gene of the banana aphid was targeted for silencing by RNAi through transgenic expression of *AChE* dsRNA in banana and plantain plants. The efficacy of dsRNA was first assessed using an artificial feeding assay. In vitro aphid feeding on a diet containing 7.5% sucrose, and sulfate complexes of trace metals supported aphid growth and reproduction. When *AChE* dsRNA was included in the diet, a dose of 500 ng/μL was lethal to the aphids. Transgenic banana cv. Cavendish Williams and plantain cvs. Gonja Manjaya and Orishele expressing *AChE* dsRNA were regenerated and assessed for transgene integration and copy number. When aphids were maintained on elite transgenic events, there was a 67.8%, 46.7%, and 75.6% reduction in aphid populations growing on Cavendish Williams, Gonja Manjaya, and Orishele cultivars, respectively, compared to those raised on nontransgenic control plants. These results suggest that RNAi targeting an essential aphid gene could be a useful means of reducing both aphid infestation and potentially the spread of the disease they transmit.

## 1. Introduction

Banana bunchy top disease (BBTD) is one of the most economically important diseases affecting banana and plantain (*Musa* spp.) production [1,2]. BBTD is caused by the banana bunchy top virus (BBTV) and is transmitted within short distances by banana aphids (*Pentalonia nigronervosa* Coquerel, superfamily Aphididae, order Hemiptera) or long distances through infected planting materials [3]. In severe cases, the disease can cause up to 100% yield loss [4]. Excessive use of insecticides to control the spread of BBTD by aphids is not ideal due to environmental risks, the potential for adverse effects to off-target organisms and long-term acquired insecticide resistance [5]. Breeding for natural resistance to the banana aphid would be one approach of curtailing the disease. However, there are no banana/plantain germplasms with resistance to aphids, and moreover, breeding of banana is both time-consuming and challenging. Molecular breeding offers an alternative approach for crop improvement. The availability of robust and reproducible transformation systems for banana have made it possible to genetically modify this crop for enhanced agronomic traits [6,7,8,9]. The development of aphid-resistant banana and plantain cultivars through genetic modification could provide a durable strategy towards controlling BBTD.

RNA interference (RNAi) is a post-transcriptional gene silencing mechanism that has proven to be effective against some insect pests, including the coleopteran, lepidopteran, and hemipteran orders [10,11,12,13,14,15]. These studies showed that the insecticidal effectiveness of RNAi is strongly dependent on the selected target gene and other factors such as dsRNA fragment length and the relative susceptibilities of different insect developmental stages. For example, of the nine genes targeted against *Myzus persicae* (green peach aphid), dsRNAs specific for the *V-ATPaseE* and *TBCD* genes were the most effective, while expression of an artificial micro (ami) RNA targeting the *AChE2* gene provided greater aphid resistance compared to hairpin (hp) RNA-expressing plants [16]. Likewise, targeting the E-subunit of the *V-ATPase* gene resulted in suppression of a broader range of insects, including *Tribolium castaneum*, *Manduca sexta*, *Acyrthosiphon pisum*, and *Drosophila melanogaster* [17]. In addition, planthoppers (*Nilaparvata lugens)* fed on transgenic rice plants expressing dsRNA against the *NlHT1*, *Nlcar* and *Nltry* genes showed reduced target transcript levels in their midgut, but no lethality was observed [11].

Acetylcholinesterase (AChE) is an enzyme that catalyzes the breakdown of acetylcholine and plays a significant role in the physiology and survival of insects [16]. Anticholinesterase compounds such as organophosphate, carbamate, chlorpyrifos, and methamidophos phosphorylate [18,19,20,21] are commonly employed as insecticides to control agricultural insects and pests [18,19,20,21]. Blocking *AChE* function can cause mortality, malformation, and/or inhibit adequate growth.

An artificial diet is an essential step for developing a noninvasive tool for gene functionality studies and validating candidate target genes for RNAi-mediated control of aphids. To date, no artificial diet has been developed for banana aphid, so in this study we developed a well-suited synthetic diet by optimizing existing diets. The optimized diet was further used to study the effect of dsRNA-*AChE* ingestion on banana aphids under a controlled condition.

In this study, we report the effect of *AChE* dsRNA uptake on the banana aphid, *Pentalonia nigronervosa*. To do this, we first developed an artificial liquid feeding complex containing sucrose and increasing levels of *AChE* dsRNA. Aphids fed on the dsRNA-containing media were monitored over time for mortality. Transgenic banana and plantain expressing hpRNA targeting the *AChE* gene were then generated by *Agrobacterium*-mediated transformation of cell suspensions. Plants were genetically characterized, and elite events inoculated with aphids in a controlled environment chamber. Aphid populations were greatly reduced on some transgenic events compared to those on the control nontransgenic plants.

## 2. Results and Discussion

### 2.1. Optimization of an In Vitro Feeding Diet for Banana Aphids

Four diet formulas (Diets 1–4) and water as control (Diet 5), containing increasing levels of sucrose, were assessed for their capacity to support aphid growth over five and seven days (Table 1). Diets 1, 2, 3, and 4 supported aphid growth, with the highest mean aphid and nymph populations observed on Diet 2 containing sucrose at a level of 7.5 g/100 mL at day 5 (mean number of aphids = 3.83 ± 0.5; mean number of nymphs = 8.44 ± 1.8) and day 7 (mean number of aphid = 4.0 ± 0; mean number of nymph = 9.8 ± 1.2). Sucrose levels between 5 and 20 g/100 mL were most suitable to support the survival of banana aphids. Highest aphid mortality was observed in Diets 1–4 containing no sugar or very high sucrose (30 g/100 mL) and in the control Diet 5 due to the lack of essential nutrients. The intrinsic rate of natural increase and finite rate was highest in Diet 2 at 7.5 g/100 mL sucrose concentration (Figure 1). These results showed that diet composition and sucrose levels are important factors for aphid growth and that a diet with balanced carbohydrates and amino acids is likely essential for aphid survival and reproduction [22]. Furthermore, trace metals like iron, zinc, manganese, and copper are vital for aphid reproduction [23]. The availability of some of these minerals either in chelated (Diet 1) or sulfate (Diet 2) forms supported the development and reproduction of banana aphids. In addition, Diets 3 and 4, which contained sulfate and chloride complexes without the inclusion of trace metals, supported the growth of banana aphids for the period tested. This could be due to the availability of magnesium as well as amino acid-containing compounds in the diet. A study on pea aphids [24,25] confirmed that the use of trace elements (Fe, Cu, Mn, and Zn) in sequestrate forms, improved the viability of these aphids. In this study, sulfate complexes were used for all the trace metals in Diet 2 due to the unavailability of a sequestered form of manganese. Despite this, aphid growth and reproduction did not appear to be compromised. However, an extended period of feeding is required to further confirm this finding.

### 2.2. Dose Effects of dsRNA-AChE on Survival of Banana Aphids

Synthetic Diet 2 (containing 7.5 g/100 mL sucrose), identified as a suitable diet composition for banana aphid feeding and growth, was supplemented with dsRNA-*AChE* at increasing concentrations of 0, 100, 200, 300, and 500 ng/μL to determine its effects on aphid survival. By monitoring aphid numbers over 2, 3, and 7 days of feeding, increasing concentrations of dsRNA-*AChE* were shown to significantly impact aphid survival, with a dose of 500 ng/μL of dsRNA having the highest lethality (Table 2, Figure 2).

*AChE* is an essential enzyme in the aphid life cycle [16] and is known to activate the breakdown of acetylcholine (ACh) through hydrolysis, thereby preventing possible adverse effects associated with overaccumulation of an impulse generated by this neurotransmitter [26]. Choline and acetate ions are the byproducts of the reaction. Choline reinitiates the production of *AChE,* while acetate ion plays a vital role in the synthesis of fatty acids and carbohydrates when it combines with coenzyme A [27]. *AChE,* therefore, represents a key enzyme target to control crop pests like aphids [28].

The optimum lethal dose of 500 ng/μL dsRNA-*AChE* for *Pentalonia nigronervosa* in this study is about three-fold higher than that for the Asian citrus psyllid (*Diaphorina citri*) (125 ng/μL dsRNA-*AChE*) [28], but two-fold lower than that for the *BC-Actin* gene (1000 ng/μL) of *Bactericera cockerelli* that was required to reduce gene expression levels [29]. Other studies showed that an injection delivery procedure of 5 μg dsRNA-*AChE* in *Tuta absoluta* caused 63% mortality [30].

### 2.3. Generation and Molecular Characterization of Transgenic Events

In total, 80 putative transgenic plants were generated on selective regeneration media supplemented with kanamycin (100 mg/L) (Figure 3) following *Agrobacterium*-mediated transformation of embryogenic cell suspensions of Cavendish Williams, Orishele, and Gonja Manjaya. PCR using primers designed to amplify a 466 bp fragment of the *AChE* hairpin (Figure 4) showed the presence of the transgene in 79 events out of 80 transgenic events tested. The integration and copy number of the transgene was determined by Southern blot analysis using a digoxigenin (DIG)-labeled probe specific for the *AChE* gene fragment. Based on unique hybridization signals, all plants were likely independent transgenic events. Of the selected PCR positive events analyzed by Southern hybridization, total of 4, 8, and 11 events of Cavendish Williams, Orishele, and Gonja Manjaya showed a single copy number of transgene, respectively (Figure 5). Further, RT-PCR analysis was performed to confirm the expression of the *AChE* transgene in a subset of transgenic plants. Although not a quantitative assay, the intensity of RT-PCR amplicons following agarose gel electrophoresis suggested that these events varied in dsRNA*-AChE* expression levels with only one event of Gonja Manjaya and three events of Orishele having undetectable levels (Figure 6).

### 2.4. Evaluation of Transgenic Events for Resistance to Aphids

In total, 71 transgenic events were challenged with *P. nigronervosa* inside insect-proof cages within a conviron plant growth chamber to determine their resistance levels to the banana aphids. Five first instar aphids were placed on the pseudostem of three biological replicates of each transgenic event, and the aphid population was monitored every 7 days for 21 days. The relative decrease of banana aphid populations in comparison to the nontransgenic control was used as a measure of “relative resistance”. The results showed that the transgenic banana and plantain events had varied levels of resistance to the aphids compared to control plants (Figure 7). For transgenic Cavendish Williams, four transgenic events showed an increase in the aphid population (−0.3 to −67.8%) relative to the control, while 28 events showed a decrease in the aphid population ranging from 7.7% to 79%. In the case of Orishele, two events had a higher aphid population than the control (−5.8% and −10%), while eight events showed a decrease in the aphid population (ranging from 22.6% to 75.6%). For transgenic Gonja Manjaya, 14 out of 29 events tested, showed a decrease in aphid populations relative to the control plants, ranging between 0.7% and 46.7%. However, the remaining 15 events of Gonja Manjaya showed an increase in aphid population ranging from −4.8% to −65.9% relative to the controls. A comparison between nontransgenic control plants, Agbagba, Obino l’Ewai, Orishele, Gonja Manjaya, and Cavendish Williams, confirmed that there was no significant difference in aphid populations upon feeding across these cultivars (Figure 8).

Of the cultivars tested, expression of dsRNA-*AChE* appeared most effective in Orishele and Cavendish Williams, with 80% and 87.5% of events showing a reduction in aphid populations compared to the controls, respectively. In contrast, only 48% of Gonja Manjaya events showed reduced aphid numbers. This is unlikely the result of cultivar preference by the aphid as there was no significant difference in mean aphid populations over time across nontransgenic plants representing these cultivars. Although there seems not to be a direct relationship between the differential expression levels and the variation in resistance in the current study, the variation in resistance levels between individual events is most likely linked to the differential expression levels of dsRNA-*AChE* and the stoichiometric ratio of small interfering RNAs (siRNA) accumulation, the initiators of RNAi, and the resultant sequence-specific degradation of target gene mRNA. In this study, dsRNA-*AChE* was nuclear-expressed and, therefore, the long *AChE*-dsRNA is likely processed into siRNAs within the cytoplasm by Dicer before ingestion by the aphid. The siRNAs are known to be less effective at stimulating RNAi than long dsRNA templates [31]. This factor, together with other aspects such as optimal target sequence and long RNA and siRNA stability may contribute to the level of resistance and the variation in resistance observed. It was previously reported that the pH of an insect or midgut secretion can degrade dsRNA [32,33] and reduce the downstream systemic RNAi effect. Peng et al. [34] pointed out that depending on the insect species, the pH in the gut favors the activity of some nucleases, which could be more aggressive than others in degrading dsRNA. For instance, the high pH in the gut of some Leptidoptan species (e.g., *Spodoptera litura*) has been associated with high nucleolytic degradation of dsRNA by nucleases present there. In a study by Guo et al. [16], engineered aphid resistance was less effective when expressing hairpin RNA targeting the *MpAChE2* gene compared to amiRNAs targeting two *MpAChE2* sequences. Perhaps a similar amiRNA strategy would be equally effective against *P. nigronervosa* in transgenic banana and plantain. Alternatively, chloroplast-based expression of *AChE* dsRNA would sequester the long dsRNA from Dicer and deliver a more potent elicitor of RNAi into the aphid. Other strategies such as topical application of transdermal dsRNA delivery system using nanotechnology targeting important aphid genes have a potential of managing aphid infestation in the field [35]. However, this approach warrants investigation for its efficiency in controlling aphid population and transmission of virus during probing.

## 3. Materials and Methods

### 3.1. Source of Plant Material

One plantain cultivar Orishele, commonly grown in West Africa and preferred by farmers in that region, was obtained from the Genetic Resources Center of the International Institute of Tropical Agriculture (IITA), Nigeria. Onother plantain cultivar Gonja Manjaya, preferred by East African farmers, and one dessert banana cultivar Cavendish Williams, were obtained from the Plant Transformation Laboratory of IITA, Kenya.

### 3.2. Rearing and Maintenance of Aphids in the Laboratory

A single banana aphid (*P. nigronervosa*) was isolated from a banana plant in the field (GPS coordinates: latitude −1.270224°, longitude 36.72312°, altitude 1823.00 m) at the International Livestock Research Institute (ILRI) Nairobi, Kenya. The aphid was maintained on an eight-week-old potted banana plant in an insect-proof cage under natural light and temperature. After colony establishment, a single aphid was isolated from the plant and used to initiate a fresh population on a young banana plant. A pure colony was maintained by transferring the aphids to newly acclimatized potted banana plants every 4–8 weeks.

### 3.3. Optimization of an Artificial Aphid Feeding Medium

*P. nigronervosa* diet compositions were prepared based on ingredients described by Douglas and van Emden [36]. The predominant modifications were in the metal complexes in place of sequestered manganese (Table 3) and the omission of riboflavin. In addition, increasing concentrations of sucrose (0, 5, 7.5, 15, 20, and 30 g per 100 mL) were included for each diet type (Diet 1, Diet 2, Diet 3, Diet 4, and control Diet 5 containing Milli-Q water only). The pH of each diet was adjusted to 7.5 with 1 M KOH and filter-sterilized using a 0.2 μm filter in a laminar flow hood. Diet aliquots were stored at −20 °C and thawed immediately prior to use.

### 3.4. Construction of Plasmids for In Vitro dsRNA Synthesis and Plant Transformation

#### 3.4.1. Amplification and Sequencing of a *P. nigronervosa* Partial *AChE* Gene Fragment

*AChE* gene sequences from related insect species were extracted from GenBank and aligned. These sequences included accessions AB158639, JQ349160, KJ755948, AY819704, KJ561353, and XM_008187580. Based on regions of homology, two primers, ACE-F1 (5′-GCTACTTAGTAGCTTCTACTTATGGACTTTC-3′) and ACE-R1 (5′-CCCAAGGTGCTGTTGAGGATCCTGATTCCATGA-3′) were designed to amplify approximately 1100 bp fragment of the *AChE* gene from *P. nigronervosa* genome. Total DNA was extracted from approximately ten aphids collected from three geographical regions (including Australia, Kenya, and Cameroon) using the CTAB protocol of Stewart and Via [37]. Approximately 1 μg of DNA was used as the template in a PCR containing GoTaq polymerase and primer pair ACE-F1 and ACE-R1. PCR cycles were 95 °C for 30 s, followed by 30 cycles of 95 °C for 30 s, 50 °C for 30 s, 72 °C for 45 s, and a final extension of 72 °C for 2 min. PCR amplicons were gel-purified and cloned into pGEM^®^-T Easy (Promega) vector according to the manufacturer’s guidelines. Two clones representing each geographical location were Sanger sequenced and aligned. From this, a 100% conserved region of approximately 400 bp mapping to exon 3 of the *AChE* coding region was selected for RNAi construct development. 

#### 3.4.2. Cloning of *P. nigronervosa Ache* Gene Target Sequence

Two primers, ACE400-F (5′-GATCTTAGATTTAGGCACCCTCGCCCAATTG-3′) and ACE400-R (5′-AAGTCCAGCGTTCCCTGGAACATCTTC-3′) were used to amplify the 400 bp *AChE* target sequence from *P. nigronervosa* total DNA. PCR was performed as described above and the amplicon similarly cloned into pGEM^®^-T Easy vector and sequenced. The resulting plasmid named pGEMT-ACE400 was used as the template for in vitro dsRNA synthesis. 

#### 3.4.3. Assembly of a Binary Vector Capable of Expressing *P. nigronervosa AChE* dsRNA

Sense and antisense sequences of the *P. nigronervosa AChE* 400 bp gene target were amplified with primer pair ACE400-KpnI (5′-GTACCGATCTTAGATTTAGGCACCCTCGCCCAATTG-3′) and ACE400-SacI (5′-GAGCTCAAGTCCAGCGTTCCCTGGAACATCTTC-3′), and primer pair ACE400-AsiSI (5′–GCGATCGCAAGTCCAGCGTTCCCTGGAACATCTTC-3′) and ACE400-BglII (5′-AGATCTTAGATTTAGGCACCCTCGCCCAATTG-3′), respectively. PCRs were performed as described above using pGEMT-ACE400 as the template and the amplicons cloned into pGEM^®^-T Easy vector and sequenced. The antisense and the sense fragments were excised from pGEM^®^-T Easy vector and ligated into the *AsiS*I and *Bgl*II and *Kpn*I and *Sac*I sites respectively in the binary vector pNXT-TYMPhp to yield pNXT-35S-ACE-hp. The sense and antisense *AChE* fragments were separated by approximately 100 bp synthetic intron [38], to ensure stability of the hairpin. In this construct, the *AChE* gene is driven by the cauliflower mosaic virus (CaMV) 35S promoter. The plasmid also contains *npt*II gene encoding resistance to the antibiotic kanamycin for selection of transgenic plants (Figure 9). The plasmid was mobilized to *Agrobacterium tumefaciens* (strain EHA 105) by electroporation and used for banana transformation.

### 3.5. In Vitro Synthesis of dsRNA-AChE

The pGEMT-ACE400 plasmid DNA was isolated from *E. coli* using an Invitrogen PureLink Quick Plasmid Miniprep Kit (K2100-10 and K2100-11). Plasmid DNA was linearized overnight at 37 °C by restriction digestion with *Nco*l or *Sal*l to generate templates for SP6 (sense) or T7 (antisense) in vitro transcription, respectively. Single-stranded RNA was produced using SP6 and T7 RNA polymerase (Promega, Madison, WI, USA) according to the manufacturer’s specifications. Equal quantities of sense and antisense ssRNA were annealed by denaturation at 95 °C for 2 min with a stepwise reduction of 1 °C every 2 min to an end temperature of 4 °C: ∞. The final concentration of the dsRNA was determined by spectrophotometry.

### 3.6. Development of an Artificial Aphid Feeding Assay and dsRNA-AChE Dose Response

Preparation of the diet sachets and cages was done as described by Douglas and van Emden [36] using a 15 mL falcon tube cut into 10 mm × 10 mm open ends. Rough edges were smoothed to avoid piercing the parafilm, which could cause diet leakage or allow the aphids to escape.

Four aphids were placed into each diet tube and sealed appropriately. Tubes were incubated in a controlled environment chamber (CMP 6050, Canada) at 25 °C, 75% RH and 12:12 h photoperiod and aphid numbers monitored over 5 and 7 day periods. Two independent experiments using three replicates with four aphids each per diet (*n* = 24) were performed to determine the appropriate diet for in vitro rearing of banana aphids.

For dsRNA dose-response assays, four aphids were placed in diet tubes containing Diet 2 formula (with 7.5% *w/v* sucrose) and increasing concentrations (0, 100, 200, 300, and 500 ng/μL) of in vitro synthesized *AChE*-dsRNA. Two independent experiments using three replicate with four aphids each per diet dose (*n* = 24) were performed at 2, 3, and 7 days post-feeding.

### 3.7. Agrobacterium-Mediated Transformation of Plantain and Banana and Generation of Transgenic Events

Embryogenic cell suspension of banana cultivar, Cavendish Williams and plantain cultivars, Gonja Manjaya and Orishele were transformed using *Agrobacterium tumefaceins* strain EHA 105 carrying the binary vector pNXT-ACE-35S-hp and plants regenerated as described by Tripathi et al. [7]. Transgenic events were regenerated on selective media containing kanamycin (100 mg/L).

### 3.8. Molecular Characterization of Transgenic Plants

#### 3.8.1. Genomic DNA Extraction and Validation of Transgenic Events by PCR Analysis

Genomic DNA was isolated from each putative transgenic event using a modified CTAB protocol as described by Tripathi et al. [7]. Presence of the transgene sequence confirmed by PCR using a forward *AChE* specific (F: GAGCTCAAGTCCAGCGTTCCCTGGA) and a reverse primer specific to the syntron region (R: AGAATTGGCGCGCCATTTAAATC) designed to amplify a product of 466 bp.

#### 3.8.2. Southern Blot Analysis to Confirm the Integration of Transgene

About 20 μg of genomic DNA was mixed with 100 U of *Nco*I restriction enzyme HF (R3193S) and digested at 37 °C for 18 h. Digests (which included transgenic plants, nontransgenic control as negative control, and *AChE* plasmid DNA as positive control) were concentrated and electrophoresed through 0.8% (*w/v*) agarose gel and transferred to a positively charged nylon membrane. The membrane was hybridized with a DIG-labeled probe specific to the *AChE*-400 fragment sequence and constituted according to manufacturer’s instructions. The labeled probe and unlabeled reaction mix were incubated in a thermocycler at 95 °C for 5 min, (95 °C for 30 s, 60 °C for 30 s, and 72 °C for 1 min: 34 cycles), 72 °C for 2 min and 4 °C: ∞). About 10 μL of the labeled probe was added to 40 μL of nuclease-free water, denatured in boiling water for 10 min, and placed immediately on ice for 5 min. The probe was hybridized to the membrane for 18 h at 42 °C and 60 rpm, the membrane washed, and signal detected using CDP-Star chemiluminescent substrate. 

#### 3.8.3. RT-PCR Analysis to Confirm Expression of AChE-dsRNA in Transgenic Plants

RNA was extracted from plant leaves using a Qiagen RNeasy Plus Mini Kit (catalog number 74134) and cDNA synthesized using a Maxima reverse transcriptase enzyme mix (Thermo Scientific, K1642 and R1362). RT-PCRs were performed using the following primer pairs: *AChE* specific primers (F: CCCTGGAACATCTTCAGTG; R: TAGGCACCCTCGCCCAATTG; product size: 376 bp) and *Musa* 25S housekeeping gene-specific primers (F: ACATTGTCAGGTGGGGAGTT; R: CCTTTTGTTCCACACGAGATT; product size: 106 bp). The reaction mix was incubated at 95 °C for 15 min, (94 °C for 30 s, 57 °C for 30 s, 72 °C for 30: 35 cycles), 72 °C for 5 min and storage 4 °C.

#### 3.8.4. Evaluation of Transgenic Events for Resistance to Aphids

Three biological replicates of each transgenic event and nontransgenic control plants were potted into a mixture of sterile soil and chicken manure in the ratio of 3:1 and acclimatized in a greenhouse. Plants were irrigated with water and placed in a humidity chamber at 16/8 h photoperiod, 25 °C ± 2.0 and watered sparingly every week for about 8 weeks.

Plants were placed inside an insect-proof cage and five first instar aphids inoculated onto the pseudostem of each plant. The cage was then placed inside a controlled Conviron growth chamber at 25 °C, 75% relative humidity with a 12 h photoperiod (Figure 10). The population of aphids was counted every 7 days after inoculation for 21 days. This experiment was repeated three times for statistical relevance.

### 3.9. Statistical Analysis

The statistical analysis was done using the generalized linear model (PROC GLM) using the Statistical Analysis System (SAS V 9.4, Cary, NC, USA) to do the analysis of variance (ANOVA) so as to obtain the variance components and least square means (LSMEANS). The Dunnett’s test was to compare the treatments to their respective controls. The percentage relative resistance was estimated based on aphid numbers on transgenic events compared to aphid numbers on control plants using the formula
X (%) = [(a − b) ÷ a] × 100
where X is relative resistance or percentage decrease, a is average aphid population on control plants, and b is average aphid population on transgenic plant.

## 4. Conclusions

RNAi is a convenient means of engineering resistance to plant pests and diseases. *Pentalonia nigronervosa* is a sap-sucking aphid and the sole insect vector of BBTV, the causal agent of the destructive bunchy top disease affecting bananas and plantains. In this study, we targeted the *AChE* gene of *P. nigronervosa* using dsRNA to elicit an RNAi response with a lethal effect to the insect. The delivery of *AChE* long dsRNA in vitro within an artificial aphid feeding medium was detrimental to aphid growth and reproduction, with a dose of 500 ng/μL being most lethal. Banana and plantains engineered to express the same *AChE*-dsRNA showed varying levels of resistance to the aphid as determined by a reduction in aphid populations compared to control nontransgenic plants. The best of these events showed a 67.8% (cv. Cavendish Williams) and 75.6% (cv. Orishele) reduction in aphid populations over a period of 21 days. To the best of our knowledge, this is the first report of transgenic banana and plantain with enhanced resistance to *P. nigronervosa*. These elite transgenic events represent a significant step forward in the fight against BBTV; however, only with long-term field assessment can their resistance to both aphid and virus be thoroughly evaluated.

## Figures and Tables

**Figure 1 plants-10-00613-f001:**
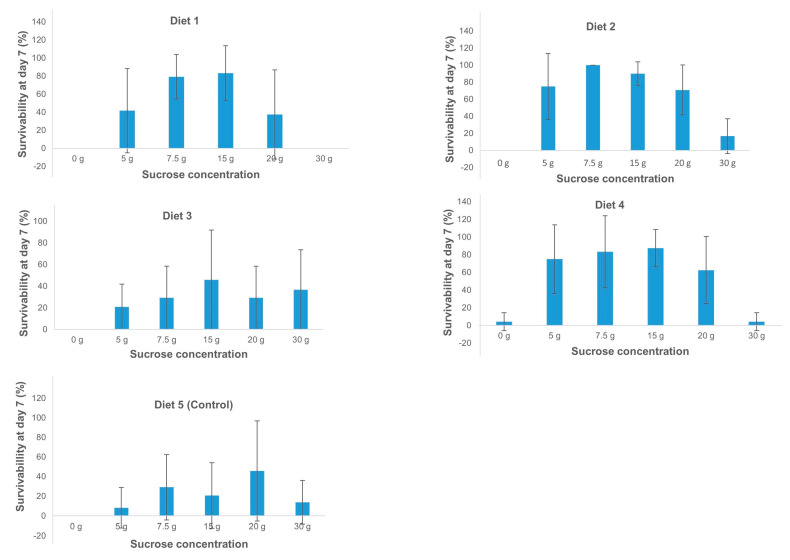
Survivability of banana aphids at day 7 post-feeding on various diets supplemented with different concentrations of sucrose. Data are presented as mean percentage survivability (%) and error lines indicate the standard deviation.

**Figure 2 plants-10-00613-f002:**
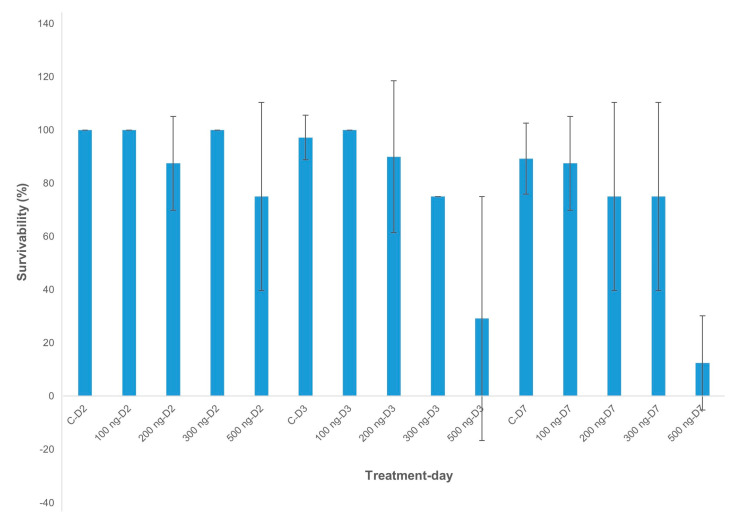
Banana aphid survivability curve at varying dsRNA concentrations and time intervals at day 2 (D2), day 3 (D3), and day 7 (D7). Data are presented as mean percentage survivability (%) and error lines indicate the standard deviation.

**Figure 3 plants-10-00613-f003:**
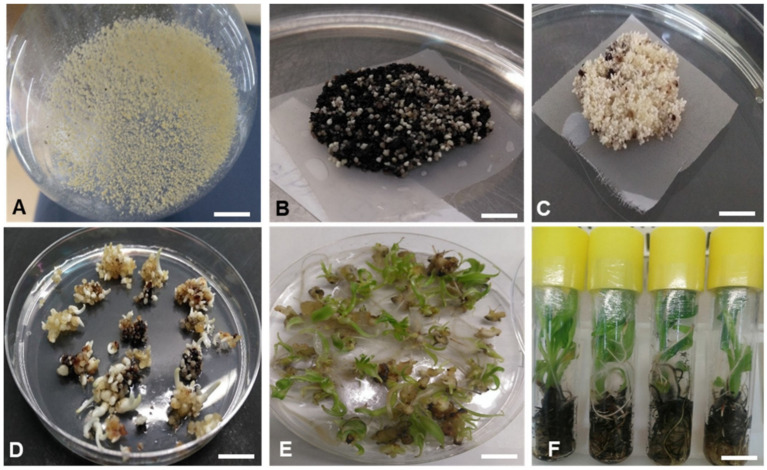
Transformation, selection, and regeneration of putative transgenic events. (**A**) Proliferating embryogenic cell suspension, (**B**) developing embryos on transformed embryogenic cells on selection medium, (**C**) developing embryos on control untransformed embryogenic cells on nonselective medium, (**D**,**E**) germinating embryos, (**F**) fully regenerated putative transgenic plant. Scale bar = 1 cm.

**Figure 4 plants-10-00613-f004:**
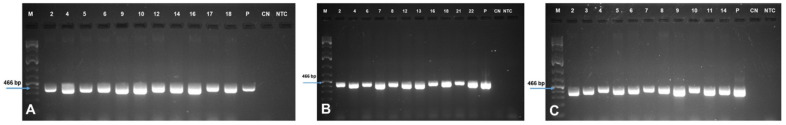
PCR amplification using *AChE* specific primers to show the presence of transgene in the transgenic banana and plantain events: (**A**) Cavendish Williams, (**B**) Orishele, (**C**) Gonja Manjaya. M—molecular marker 1 kb ladder; NTC—nontemplate control; CN—control nontransgenic plant; +—plasmid as positive control, numbers on lanes of the image (**A**–**C**) indicate the independent transgenic events of Cavendish Williams, Orishele, and Gonja Manjaya, respectively.

**Figure 5 plants-10-00613-f005:**
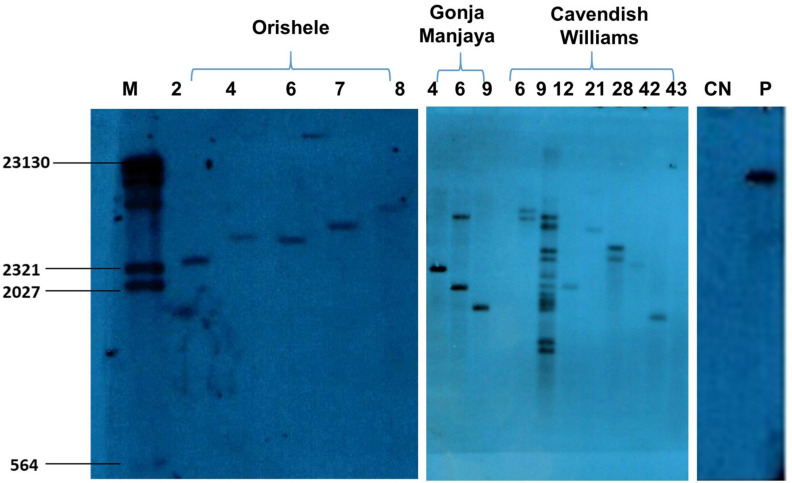
Southern blot analysis to confirm the integration of *AChE* transgene in selected transgenic events representing plantain (Orishele and Gonja Manjaya) and banana (Cavendish Williams) cultivars. M—molecular marker; CN—control nontransgenic plant; P—plasmid DNA as positive control.

**Figure 6 plants-10-00613-f006:**
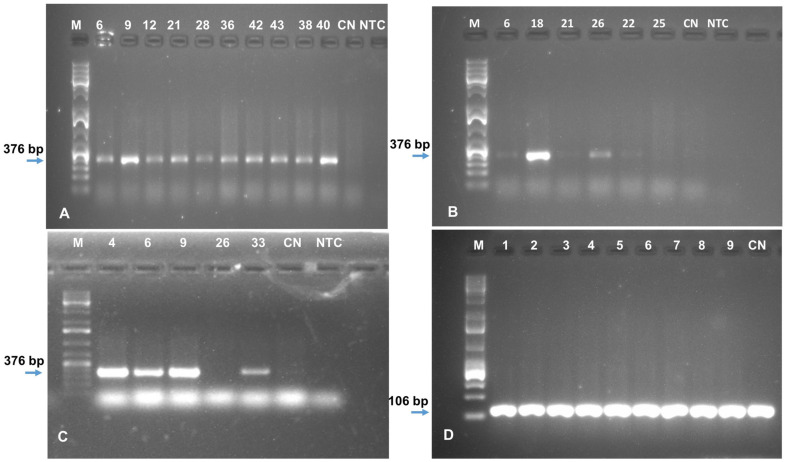
RT-PCR analysis to confirm expression of dsRNA-*AChE* in selected transgenic events. (**A**) Cavendish Williams, (**B**) Orishele, (**C**) Gonja Manjaya, (**D**) *Musa* 25S housekeeping gene in selected transgenic plants as an internal control. M—molecular marker 1 kb ladder; CN—control nontransgenic plant; NTC—nontemplate control.

**Figure 7 plants-10-00613-f007:**
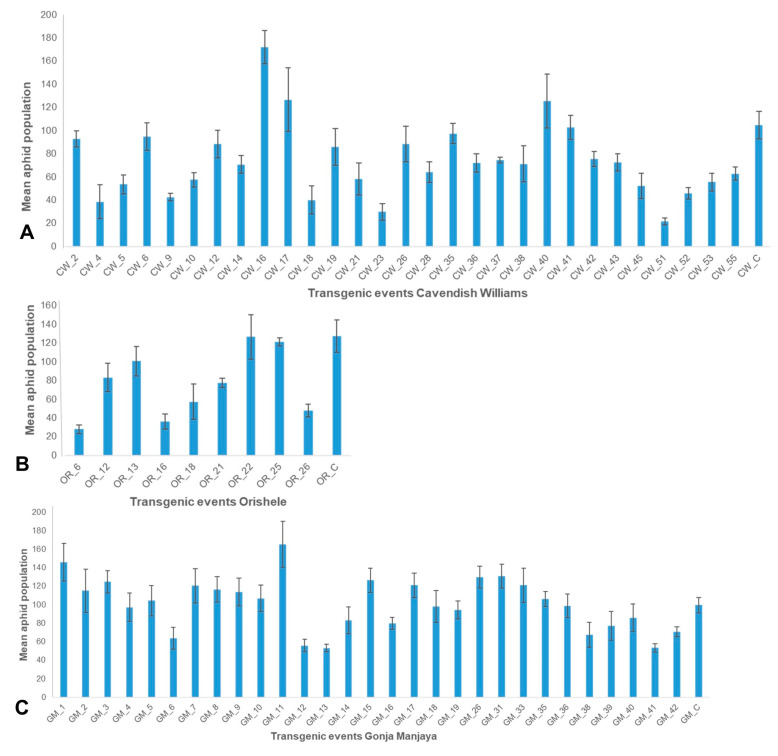
Relative change in aphid populations on dsRNA-*AChE* transgenic plants compared to nontransgenic control plants upon challenge with aphids. (**A**) Cavendish Williams, (**B**) Orishele, (**C**) Gonja Manjaya. The different number of transgenic events indicates the independent events generated for the three cultivars tested. Data are presented as LSmean values and standard error (SE).

**Figure 8 plants-10-00613-f008:**
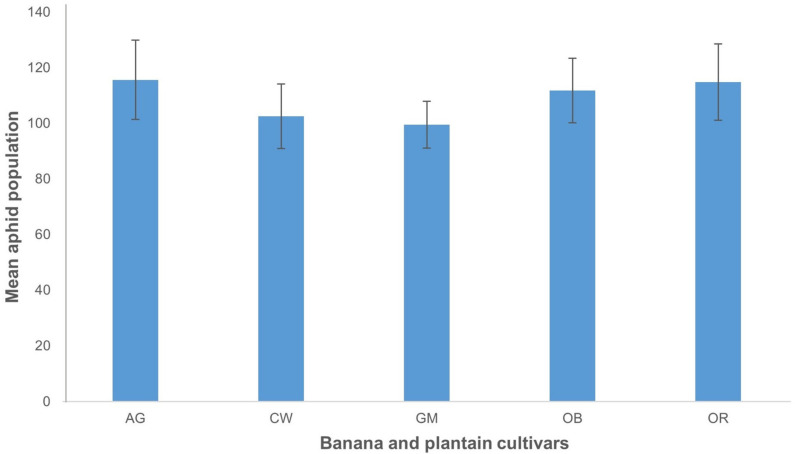
Mean live aphid population growing on nontransgenic plants of various cultivars of banana and plantain. AG—Agbagba, CW—Cavendish Williams, GM—Gonja Manjaya, OB—Obino l’Ewai, and OR—Orishele. Data are presented as means with standard error.

**Figure 9 plants-10-00613-f009:**
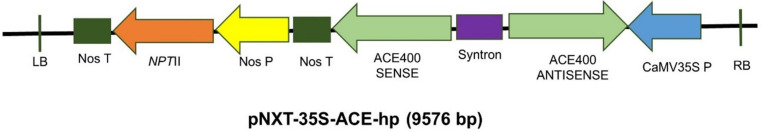
A graphical representation of the T-DNA region of the pNXT-35S-ACE hp plasmid.

**Figure 10 plants-10-00613-f010:**
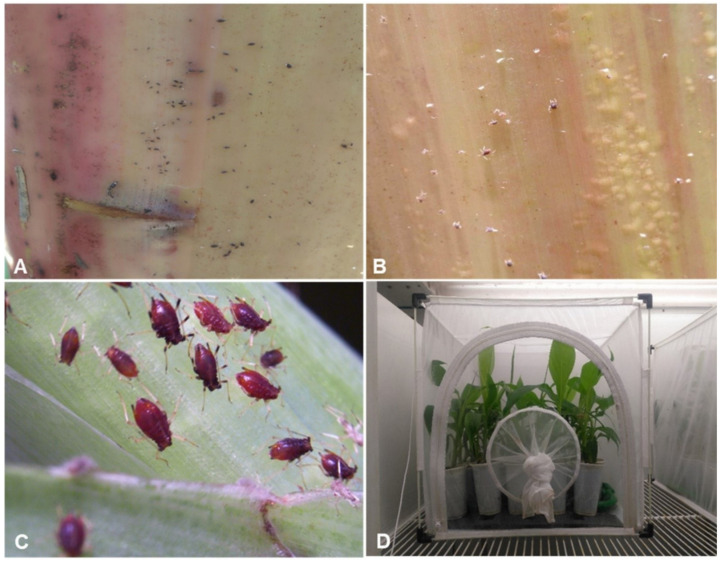
Evaluation of transgenic banana and plantains expressing *AChE*-dsRNA by challenging with aphids. (**A**–**C**) Aphids rearing on the leaves of a nontransgenic banana plant and (**D**) aphid-challenged transgenic plants in an insect-proof cage within a controlled environment chamber.

**Table 1 plants-10-00613-t001:** Effect of sucrose levels and diet type on banana aphid survival.

Day	Sucrose Level (g/100 mL)	Diet Type	NDA	NLA	NN
5	0	Diet 1	4.0 ± 0 ^ns^	0 ± 0 ^ns^	0.08 ± 0.3 ^ns^
		Diet 2	4.0 ± 0 ^ns^	0 ± 0 ^ns^	0.05 ± 0.2 ^ns^
		Diet 3	4.0 ± 0 ^ns^	0 ± 0 ^ns^	0.13 ± 0.4 ^ns^
		Diet 4	4.0 ± 0 ^ns^	0.08 ± 0.3 ^ns^	0.33 ± 0.7 ^ns^
		Control	4.0 ± 0	0 ± 0	0.08 ± 0.3
	5	Diet 1	2.0 ± 1.0 **	1.75 ± 1.6 **	3.5 ± 1.7 ***
		Diet 2	1.0 ± 1.1 ***	3.2 ± 1.2 ***	5.73 ± 2.7 ***
		Diet 3	1.0 ± 1 ***	3.13 ± 1.1 ***	3.73 ± 1.0 ***
		Diet 4	1.0 ± 1 ***	3.25 ± 1.1 ***	3.66 ± 1.8 ***
		Control	4.0 ± 1.0	0.16 ± 0.6	0.25 ± 0.5
	7.5	Diet 1	1.5 ± 1.5 ***	2.5 ± 1.5 ***	3.08 ± 1.8 **
		Diet 2	0.17 ± 0.5 ***	3.83 ± 0.5 ***	8.44 ± 1.8 ***
		Diet 3	0.71 ± 0.9 ***	3.28 ± 0.9 ***	4.21 ± 0.9 ***
		Diet 4	1.25 ± 1.4 ***	2.75 ± 1.4 ***	3.66 ± 1.9 ***
		Control	3.33 ± 1.1	0.66 ± 1.1	0.83 ± 1.3
	15	Diet 1	1.5 ± 1.7 *	2.5 ± 1.7 *	3.08 ± 1.9 **
		Diet 2	0.78 ± 0.7 ***	3.21 ± 0.7 ***	5.35 ± 2.2 ***
		Diet 3	1.2 ± 1.5 **	2.8 ± 1.5 **	3.26 ± 1.6 ***
		Diet 4	0.66 ± 0.8 ***	3.33 ± 0.8 ***	3.0 ± 1.6 **
		Control	2.8 ± 1.3	1.16 ± 1.3	0.75 ± 1.1
	20	Diet 1	2.3 ± 1.9 ^ns^	1.67 ± 1.9 ^ns^	2.16 ± 1.9 ^ns^
		Diet 2	1.4 ± 1.2 ^ns^	2.6 ± 1.2 ^ns^	3.0 ± 1.8 ***
		Diet 3	1.33 ± 1.3 ^ns^	2.66 ± 1.3 ^ns^	2.53 ± 1.5 **
		Diet 4	1.16 ± 1.3 *	2.83 ± 1.3 *	2.58 ± 0.7 **
		Control	2.66 ± 1.6	1.33 ± 1.6	0.75 ± 0.9
	30	Diet 1	3.08 ± 1.7 ^ns^	0.83 ± 1.6 ^ns^	1.08 ± 1.3 ^ns^
		Diet 2	3.13 ± 1.12 ^ns^	0.86 ± 1.1 ^ns^	0.46 ± 0.8 ^ns^
		Diet 3	3.00 ± 1.5 ^ns^	0.86 ± 1.2 ^ns^	0.86 ± 1.3 ^ns^
		Diet 4	3.66 ± 0.5 ^ns^	0.33 ± 0.5 ^ns^	0.42 ± 0.7 ^ns^
		Control	3.58 ± 0.8	0.42 ± 0.8	0.25 ± 0.5
7	0	Diet 1	4.0 ± 0 ^ns^	0 ± 0 ^ns^	0 ± 0 ^ns^
		Diet 2	4.0 ± 0 ^ns^	0.0 ± 0 ^ns^	0 ± 0 ^ns^
		Diet 3	4.0 ± 0 ^ns^	0 ± 0 ^ns^	0.33 ± 0.5 ^ns^
		Diet 4	3.83 ± 0.4 ^ns^	0.16 ± 0.4 ^ns^	0.66 ± 0.8 *
		Control	4.0 ± 0	0 ± 0	0 ± 0
	5	Diet 1	2.5 ± 1.5 ^ns^	1.66 ± 1.9 ^ns^	3.0 ± 2.1 ^ns^
		Diet 2	1.0 ± 1.5 **	3.0 ± 1.5 **	6 ± 3.7 ***
		Diet 3	0.5 ± 0.8 ***	3.5 ± 0.8 ***	4.16 ± 0.8 **
		Diet 4	1.0 ± 1.5 **	3.0 ± 1.5 **	4.0 ± 1.1 **
		Control	3.66 ± 0.8	0.33 ± 0.8	0.5 ± 0.5
	7.5	Diet 1	0.83 ± 1.0 *	3.16 ± 1.0 **	4.5 ± 1.2 **
		Diet 2	0.0 ± 0 **	4.0 ± 0 ***	9.8 ± 1.2 ***
		Diet 3	1.16 ± 1.2 ^ns^	2.83 ± 1.2 ^ns^	4.33 ± 1.0 **
		Diet 4	0.66 ± 1.6 **	3.33 ± 1.6 **	4.83 ± 2.0 ***
		Control	2.83 ± 1.3	1.16 ± 1.3	1.5 ± 1.6
	15	Diet 1	0.66 ± 1.2 **	3.33 ± 1.2 **	3.66 ± 1.6*
		Diet 2	0.4 ± 0.5**	3.6 ± 0.5 **	7.2 ± 0.8 ***
		Diet 3	2.16 ± 1.8 ^ns^	1.83 ± 1.8 ^ns^	3.66 ± 2.0*
		Diet 4	0.5 ± 0.8 **	3.5 ± 0.8 **	3.66 ± 1.9 *
		Control	3.16 ± 1.3	0.8 ± 1.3	1.17 ± 1.2
	20	Diet 1	2.5 ± 2.0 ^ns^	1.5 ± 2.0 ^ns^	3.16 ± 2.1*
		Diet 2	1.16 ± 1.2 ^ns^	2.83 ± 1.2 ^ns^	4.5 ± 2.0 ***
		Diet 3	1.83 ± 1.2 ^ns^	2.16 ± 1.2 ^ns^	3.33 ± 1.0 *
		Diet 4	1.5 ± 1.5 ^ns^	2.5 ± 1.5 ^ns^	2.66 ± 0.8 ^ns^
		Control	2.16 ± 2.0	1.83 ± 2.0	1.0 ± 0.9
	30	Diet 1	4.0 ± 0 ^ns^	0 ± 0 ^ns^	0.83 ± 1.2 ^ns^
		Diet 2	3.33 ± 0.8 ^ns^	0.66 ± 0.8 ^ns^	0.33 ± 0.8 ^ns^
		Diet 3	2.83 ± 1.5 ^ns^	1.16 ± 1.5 ^ns^	1.83 ± 1.5 *
		Diet 4	3.83 ± 0.4 ^ns^	0.16 ± 0.4 ^ns^	0.66 ± 0.8 ^ns^
		Control	3.44 ± 0.9	0.55 ± 0.9	0.33 ± 0.5
	Main effect *p*-value				
	SL		***	***	***
	DT		***	***	***
	Day		ns	ns	***
	Day*SL*DT		***	***	***

**Note:** NDA: number of dead aphids, NLA: number of live aphids, NN: number of nymphs, SL: sucrose level, DT: diet type, Diet 5: control diet. Starting point is the day of inoculation of aphid on diet = 0. The data were analyzed using the two-way ANOVA and presented as mean and standard deviation of live aphid, dead aphid, and nymph. The multiple comparison by Dunnett’s test was performed and indicated as ns, nonsignificant (*p* > 0.05); * significant (*p* ≤ 0.05); ** very significant (*p* ≤ 0.01); *** highly significant (*p* ≤ 0.001). Two independent experiments using three replicates with four aphids each per diet (*n* = 24) were performed.

**Table 2 plants-10-00613-t002:** Effect of varying concentrations of dsRNA-*AChE* on banana aphid mortality, survival, and reproduction.

Treatment (ng)	Day	NDA	NLA	NN
100	2	0 ± 0 ^ns^	4 ± 0 ^ns^	2.5 ± 0.7 ^ns^
	3	0 ± 0 ^ns^	4 ± 0 ^ns^	3.5 ± 0.7 ^ns^
	7	0.5 ± 0.7 ^ns^	3.5 ± 0.7 ***	6 ± 1.4 ^ns^
200	2	0.5 ± 0.7 ^ns^	3.5 ± 0.7 ^ns^	2.5 ± 2.1 ^ns^
	3	0.6 ± 0.9 ^ns^	3.6 ± 1.1 ^ns^	3.4 ± 1.5 ^ns^
	7	1.0 ± 1.4 ^ns^	3.0 ± 1.4 ***	7.0 ± 4.2 ^ns^
300	2	0 ± 0 ^ns^	4.0 ± 0 ^ns^	3.0 ± 1.4 ^ns^
	3	1.0 ± 0 ^ns^	3.0 ± 0 ^ns^	2.5 ± 0.7 ^ns^
	7	1.0 ± 1.4 ^ns^	3.0 ± 1.4 ***	4.5 ± 2.1 ***
500	2	1.0 ± 1.4 ^ns^	3.0 ± 1.4 ^ns^	1.5 ± 0.7 ^ns^
	3	2.83 ± 1.8 ***	1.16 ± 1.8 ***	0.66 ± 1.2 ***
	7	3.5 ± 0.7 **	0.5 ± 0.7 **	3.5 ± 2.1 *
Control	2	0 ± 0	4.0 ± 0	4.0 ± 0.7
	3	0.11 ± 0.3	3.88 ± 0.3	5.33 ± 2.5
	7	0.43 ± 0.5	3.57 ± 0.5	8.86 ± 1.9
Main effect *p*-value				
Treatment		***	***	***
Day		*	*	***
Treatment * Day		ns	ns	ns

**Note:** NDA: number of dead aphids, NLA: number of live aphids, NN: number of nymphs. Data were analyzed by two-way ANOVA to obtain the main effect and Dunnett’s test for comparison with control. Data are presented as mean and standard deviation of live aphid, dead aphid, and nymph; ns, nonsignificant (*p* > 0.05); * significant (*p* ≤ 0.05); ** very significant (*p* ≤ 0.01); *** highly significant (*p* ≤ 0.001). Two independent experiments using three replicates with four aphids each per diet (*n* = 24) were performed.

**Table 3 plants-10-00613-t003:** Diet composition for banana aphid (*Pentalonia nigronervosa*).

Composition	Quantity (mg)/100 mL Diet			
	Diet 1	Diet 2	Diet 3	Diet 4
di-potassium hydrogen orthophosphate	750.0	750.0	750.0	750.0
magnesium sulfate	-	123.0	-	123.0
magnesium chloride	123.0	-	123.0	-
L-tyrosine	40.0	40.0	40.0	40.0
L-asparagine hydrate	550.0	550.0	550.0	550.0
L-aspartic acid	140.0	140.0	140.0	140.0
L-tryptophan	80.0	80.0	80.0	80.0
L-alanine	100.0	100.0	100.0	100.0
L-arginine monohydrochloride	270.0	270.0	270.0	270.0
L-cysteine hydrochloride, hydrate	40.0	40.0	40.0	40.0
L-glutamic acid	140.0	140.0	140.0	140.0
L-glutamine	150.0	150.0	150.0	150.0
L-glycine	80.0	80.0	80.0	80.0
L-histidine	80.0	80.0	80.0	80.0
L-isoleucine (allo free)	80.0	80.0	80.0	80.0
L-Leucine	80.0	80.0	80.0	80.0
L-Lysine monohydrochloride	120.0	120.0	120.0	120.0
L-methionine	40.0	40.0	40.0	40.0
L-phenylalanine	40.0	40.0	40.0	40.0
L-proline	80.0	80.0	80.0	80.0
L-serine	80.0	80.0	80.0	80.0
L-threonine	140.0	140.0	140.0	140.0
L-valine	80.0	80.0	80.0	80.0
ascorbic acid (vitamin C)	100.0	100.0	100.0	100.0
aneurine hydrochloride (vitamin B)	2.5	2.5	2.5	2.5
nicotinic acid	10.0	10.0	10.0	10.0
folic acid	0.5	0.5	0.5	0.5
(+)-pathothenic acid (calcium salt)	5.0	5.0	5.0	5.0
myo-inositol	50.0	50.0	50.0	50.0
choline chloride	50.0	50.0	50.0	50.0
EDTA Fe(III)-Na chelate pure	1.5	-	-	-
EDTA Zn-Na_2_ chelate pure	0.8	-	-	-
MnCl_2_.4H_2_O	0.8	-	-	-
EDTA Cu-Na_2_ chelate pure	0.4	-	-	-
FeSO_4_.7H_2_O	-	1.5	-	-
ZnSO_4_.7H_2_O	-	0.8	-	-
MnSO_4_.H_2_O	-	0.8	-	-
CuSO_4_.5H_2_O	-	0.4	-	-
pyridoxine hydrochloride (vitamin B6)	2.5	2.5	2.5	2.5
D-biotin, crystalline	0.1	0.1	0.1	0.1

## Data Availability

The data presented in this study are available on request from the corresponding author.

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
