# Peer review of "Transgenic Expression of dsRNA Targeting the Pentalonia nigronervosa acetylcholinesterase Gene in Banana and Plantain Reduces Aphid Populations"

_plants, 2021, doi:10.3390/plants10040613_

Round 1

Reviewer 1 Report

It is necessary to explain what the numerical values for aphid growth in Tables 1, 2 and the result sections (2.1, 2.2) indicate (% or number of individuals?). The sample size and the number of biological replicates should also be clearly stated. The statistical analyses are also unclear as to what is compared with what kind of test; not a comparison with the control or multiple comparison? It also should be explained in the footnote of Table 1 what the starting point of Day is. There is also a lack of explanation of the motivation for the improvement of artificial feed in the Introduction section. Comparisons of population growth parameters such as a rm value (intrinsic rate of natural increase) would provide a strong evidence for artificial diet selection. It would be better to show a survival curve for the data of Table 2 to analyze the mortality effect of dsRNA-AChE. The authors tested 0-ng dsRNA as a control in Table 2, but non-target dsRNA should also be tested for the negative control. A scale bar is required for Fig. 1. Please indicate in the caption what the numbers in each lane of the gel indicate. “Of the selected PCR positive events, 4, 8 and 11 events of Cavendish Williams, Orishele and Gonja Manjaya had a single copy number, respectively (Fig. 3)” (Line 143-145). Which of “4, 8 and 11 events” should be seen in Fig. 3? Visibility in Figure 5 is extremely poor. The order of the cultivars Cavendish Williams, Gonja Manjaya and Orishele is different in each figure, which may lead to misreading. Please explain what “Mean aphid populations” indicates in Figure 6. Sections 2.4 and 2.5 of the results are missing from the manuscript. The variation in resistance levels between individual events due to “differential expression levels of dsRNA-AChE” should be more discussed using expression analysis shown in Figure 4.

Author Response

Point by point response to reviewers’ comment

We thank the reviewers for their comments and suggestions, which improve its accessibility to general readers. The manuscript has been revised based on the suggestions provided by the reviewers. All the comments and suggestions provided by the reviewers have been included in the revised manuscript.  All the changes have been marked in the revised manuscript.

Reviewer #1

  1. It is necessary to explain what the numerical values for aphid growth in Tables 1, 2 and the result sections (2.1, 2.2) indicate (% or number of individuals?).

Response: The numerical values have been explained in Table 1, and 2 and result sections.

  1. The sample size and the number of biological replicates should also be clearly stated. Response: The sample size and biological replicates have been clearly stated as recommended in the material and method section 3.6.

  1. The statistical analyses are also unclear as to what is compared with what kind of test; not a comparison with the control or multiple comparison?

Response: The statistical analyses have been defined in section 3.9

  1. It also should be explained in the footnote of Table 1 what the starting point of Day is. Response: The starting point which is the day of inoculation have been stated in the footnote of table 1.

  1. There is also a lack of explanation of the motivation for the improvement of artificial feed in the Introduction section.

Response: It has been included in the introduction section of revised manuscript.

  1. Comparisons of population growth parameters such as a rm value (intrinsic rate of natural increase) would provide a strong evidence for artificial diet selection.

Response: A graphical representation of the intrinsic rate of natural increase and finite increase have been provided as Figure 1a and 1b in the revised manuscript.

  1. It would be better to show a survival curve for the data of Table 2 to analyze the mortality effect of dsRNA-AChE.

Response:. The insect survivorship have been calculated and the survival curve ploted as Figure 2 in the revised manuscript.

  1. The authors tested 0-ng dsRNA as a control in Table 2, but non-target dsRNA should also be tested for the negative control.

Response: Thank you for your observation. It is true that we were supposed to use a non-target dsRNA as a negative control, but we did not have such a control in our lab. Based on the work of Majidiani et al., 2019 study, there was no significant difference between the two controls of dsRNA-GFP as non-target dsRNA and water as negative controls. Hence, this suggested that the dsRNA of the target genes had a significant effect on the organism tested. We therefore used water as negative control for the experiment.

  1. A scale bar is required for Fig. 1.

Response: Scale bar provided in the revised manuscript.

  1. Please indicate in the caption what the numbers in each lane of the gel indicate. “Of the selected PCR positive events, 4, 8 and 11 events of Cavendish Williams, Orishele and Gonja Manjaya had a single copy number, respectively (Fig. 3)” (Line 143-145). Response: The number on lanes of the image A, B and C indicate the independent transgenic events of Cavendish Williams, Orishele and Gonja Manjaya, respectively. It has been indicated in the figure legend.

  1. Which of “4, 8 and 11 events” should be seen in Fig. 3?

Response: 4, 8, and 11 events are the total number of events for different cultivars showing a single copy of transgene integration. It has been clarified in the revised manuscript.

  1. Visibility in Figure 5 is extremely poor.

Response: The visibility of Figure 5 (revised figure 7) has been improved.

  1. The order of the cultivars Cavendish Williams, Gonja Manjaya and Orishele is different in each figure, which may lead to misreading.

Response: The order of the figures have been rearranged as recommended.

  1. Please explain what “Mean aphid populations” indicates in Figure 6.

Response: The “mean aphid populations” is the mean of live aphid growing on non-transgenic plants of various banana and plantain. It has been revised.

  1. Sections 2.4 and 2.5 of the results are missing from the manuscript.

Response: The numbering have been corrected.

  1. The variation in resistance levels between individual events due to “differential expression levels of dsRNA-AChE” should be more discussed using expression analysis shown in Figure 4

Response: This section has been revised.

Reviewer 2 Report

Major comments

  • Patterning to the results described in 2.6 (Line 168-186), can the authors provide a supplementary file with the details of the daily survival data of the different treatments? It is currently difficult to ascertain the variability (details) in survival over the exposure period.

  • The authors started with five first instar aphids as indicated in Line 170 and Line 351-354. How long did the starting aphids survive before giving birth to live young aphids? This is because the survival of the starting aphids can have a big impact on the population analysis (at the end of the bioassay).

  • I will also suggest a figure in the material and methods that clearly shows the experimental design for the transgenic plant assays.

  • The authors should also include details on how the statistical analysis were performed in this study.

  • The statement on Line 209-210: “…It was previously reported that the pH of an insect or midgut secretion can degrade dsRNA…” should be revised (updated). Depending on the insect species, the pH in the gut favours the activity of some nucleases which could be more aggressive than others (in degrading dsRNA). For example, the high pH in the gut of some Leptidoptan species (e.g Spodoptera litura) has been associated with high nucleolytic degradation of dsRNA by nucleases present there. See reference: Peng, Yingchuan, Kangxu Wang, Wenxi Fu, Chengwang Sheng, and Zhaojun Han. "Biochemical comparison of dsRNA degrading nucleases in four different insects." Frontiers in physiology 9 (2018): 624.

  • The authors mention on Line 214-216: “…Alternatively, chloroplast-based expression of AChE dsRNA would sequester the long dsRNA from Dicer and deliver a more potent elicitor of RNAi into the aphid….”. How do they see this happen in practice, considering that aphids are phloem suckers?

  • Can the authors provide more discussion to back the statement made on Line 216 -218: “….Other strategies such as topical application of long dsRNA targeting important aphid genes warrant investigation but may be challenging to deliver under field situations….”? There are early indications of the potential of inducing RNAi effects in aphids through the use of nanocarrier-dsRNA complexes that can directly pass through the aphid cuticle (possible spray application). See: Yan, S., Qian, J., Cai, C., Ma, Z., Li, J., Yin, M., ... & Shen, J. (2020). Spray method application of transdermal dsRNA delivery system for efficient gene silencing and pest control on soybean aphid Aphis glycines. Journal of Pest Science, 93(1), 449-459.

Minor comments

Line 49: “…for the banana…”

Line 54: “…proven to be effective…”

Line105: “…Despite this, aphid growth…”

Table 1: I will suggest to round up the numbers since you cannot have a 0.3 dead aphid. For example 3.9±0.3 dead aphid to 4±0. Same for the live aphids and nymphs.

Table 2: same comment as above.

Line 241: “….of a P. nigronervosa partial…”

Line 259: “…..of P. nigronervosa AChE gene…”. Also adjust this on Line 267

Line 361-363: Can the authors provide more information to the legend of Figure 8. Is A-C showing aphids on the control or transgenic plants?

Line 369: “….effect to the insect…”

Author Response

Point by point response to reviewers’ comment

We thank the reviewers for their comments and suggestions, which improve its accessibility to general readers. The manuscript has been revised based on the suggestions provided by the reviewers. All the comments and suggestions provided by the reviewers have been included in the revised manuscript.  All the changes have been marked in the revised manuscript.

Reviewer #2

  1. Patterning to the results described in 2.6 (Line 168-186), can the authors provide a supplementary file with the details of the daily survival data of the different treatments? It is currently difficult to ascertain the variability (details) in survival over the exposure period.

Response: The data was collected at 7 day post feeding for 3 weeks. So we do not have the daily survival data.

  1. The authors started with five first instar aphids as indicated in Line 170 and Line 351-354. How long did the starting aphids survive before giving birth to live young aphids? This is because the survival of the starting aphids can have a big impact on the population analysis (at the end of the bioassay).

Response: Since data was collected only at 7 day post feeding, nymphs observed at this time was recorded. However, production of nymph was observed from 3 day post feeding (personal observation).

  1. I will also suggest a figure in the material and methods that clearly shows the experimental design for the transgenic plant assays.

Response: Please, we would appreciate if the reviewer can expatiate on this. The transgenic plants were challenged with aphids.

  1. The authors should also include details on how the statistical analysis were performed in this study.

Response: The detail of statistical analysis has been included in section 3.9.

  1. The statement on Line 209-210: “…It was previously reported that the pH of an insect or midgut secretion can degrade dsRNA…” should be revised (updated). Depending on the insect species, the pH in the gut favours the activity of some nucleases which could be more aggressive than others (in degrading dsRNA). For example, the high pH in the gut of some Leptidoptan species (e.g Spodoptera litura) has been associated with high nucleolytic degradation of dsRNA by nucleases present there. See reference: Peng, Yingchuan, Kangxu Wang, Wenxi Fu, Chengwang Sheng, and Zhaojun Han. "Biochemical comparison of dsRNA degrading nucleases in four different insects." Frontiers in physiology 9 (2018): 624.

Response: The statement has been updated as advised.

  1. The authors mention on Line 214-216: “…Alternatively, chloroplast-based expression of AChE dsRNA would sequester the long dsRNA from Dicer and deliver a more potent elicitor of RNAi into the aphid….”. How do they see this happen in practice, considering that aphids are phloem suckers?

 Response: We assume that since chloroplast do not have RNAi machinery, they do not have any dicer- or argonaute-like genes. Therefore the dsRNA remains intact in the plant for uptake by the insect and subsequent RNAi initiation in the insect. This still need to be explored.

  1. Can the authors provide more discussion to back the statement made on Line 216 -218: “….Other strategies such as topical application of long dsRNA targeting important aphid genes warrant investigation but may be challenging to deliver under field situations….”? There are early indications of the potential of inducing RNAi effects in aphids through the use of nanocarrier-dsRNA complexes that can directly pass through the aphid cuticle (possible spray application). See: Yan, S., Qian, J., Cai, C., Ma, Z., Li, J., Yin, M., ... & Shen, J. (2020). Spray method application of transdermal dsRNA delivery system for efficient gene silencing and pest control on soybean aphid Aphis glycines. Journal of Pest Science, 93(1), 449-459.

Response: This section has been revised.

Minor comments: all done

Reviewer 3 Report

The manuscript “Transgenic expression of dsRNA targeting the Pentalonia nigronervosa acetylcholinesterase gene in banana and plantain reduces aphid populations” by Temitope Jekayinoluwa, Jaindra Nath Tripathi, Benjamin Dugdale, George Obiero, Edward Muge, James Dale, and Leena Tripathi is interesting and contains valuable data for basic as well as applied science. It is a well designed and prepared study and the results seem sound and well documented.

However, to fully acknowledge the merit of the study, the following details must be included.

  1. First of all, there is no description of the statistical analysis used in the study. The results of the analysis are presented but we do not know what was compared to what, especially in the tables 1 and 2. The comparison was done among the diets at each time interval separately?
  2. Why there are different numbers of transgenic events presented in the fig. 5?
  3. The graph in the figure 6 needs a precise statement that the 0Y axis represents the number of aphids
  4. The list and sources of banana cultivars must be given in the material and methods section, and why those particular cultivars were chosen.

Author Response

Point by point response to reviewers’ comment

We thank the reviewers for their comments and suggestions, which improve its accessibility to general readers. The manuscript has been revised based on the suggestions provided by the reviewers. All the comments and suggestions provided by the reviewers have been included in the revised manuscript.  All the changes have been marked in the revised manuscript.

Reviewer #3

  1. First of all, there is no description of the statistical analysis used in the study. The results of the analysis are presented but we do not know what was compared to what, especially in the tables 1 and 2. The comparison was done among the diets at each time interval separately?

Response: The details of statistical analysis has been included in section 3.9 of revised manuscript.

  1. Why there are different numbers of transgenic events presented in the fig. 5?

Response: The transgenic events in Fig. 5 (revised figure 7) represents the total number of transgenic events tested for the three cultivars. The number of events generated for various cultivars was different.                                                                                                           

  1. The graph in the figure 6 needs a precise statement that the 0Y axis represents the number of aphids

Response: The Y axis is labelled as Mean aphid population.

  1. The list and sources of banana cultivars must be given in the material and methods section, and why those particular cultivars were chosen.

Response: Source of plant material is included in the revised manuscript.

Reviewer 4 Report

The proposed work can be published in this form. Scientifically well prepared, developped, written and exhibited. The only small proposals are to add the name of the author of the species Pentalonia nigronervosa Coquerel, as well as the order and superfamily (Hemiptera, Aphididae), because not everyone knows what an aphid is. This information can be entered on p. 1 in line 41 after (Pentalonia nigronervosa).

On a personal level, I ask a question. It has been seen how many days after the insect dies? Is this period sufficient to prevent the transmission of the virus? This is important because the species is considered to be especially harmful for BBTV transmission. However, if the aphid is infected with the virus, in the time between the acquisition of the dsRNA and its death, cannot it still transmit the virus? The bibliography agrees in stating that insecticide treatments are useless if you want to defend crops from viruses through insecticide treatments against vector insects.

Best wishes and compliments.

Author Response

Point by point response to reviewers’ comment

We thank the reviewers for their comments and suggestions, which improve its accessibility to general readers. The manuscript has been revised based on the suggestions provided by the reviewers. All the comments and suggestions provided by the reviewers have been included in the revised manuscript.  All the changes have been marked in the revised manuscript.

Reviewer #4

  1. The only small proposals are to add the name of the author of the species Pentalonia nigronervosa Coquerel, as well as the order and superfamily (Hemiptera, Aphididae), because not everyone knows what an aphid is. This information can be entered on p. 1 in line 41 after (Pentalonia nigronervosa).

Response: The information is included in the revised manuscript.

  1. On a personal level, I ask a question. It has been seen how many days after the insect dies? Is this period sufficient to prevent the transmission of the virus? This is important because the species is considered to be especially harmful for BBTV transmission. However, if the aphid is infected with the virus, in the time between the acquisition of the dsRNA and its death, cannot it still transmit the virus? The bibliography agrees in stating that insecticide treatments are useless if you want to defend crops from viruses through insecticide treatments against vector insects.

Response: BBTV acquired by aphid is likely to be transmitted within a short period of time when aphid probes into the phloem cells of an uninfected banana or plantain plant. We still need to explore if these transgenic plants will be protected with BBTV infection.

Round 2

Reviewer 1 Report

In Fig. 1a and b, it is inappropriate to connect the lines for data between the diet types. I do not understand the reason why they are separated at Day 5 and 7. Why don't you show the lx and mx curves for each diet type or sucrose content during the treatment period, and put the λ, rm, and R0 in a table? Perhaps the authors do not understand how to calculate the rm value. The survival rate per day (lx) and the number of litters (mx) must be counted until all individuals die, and the final result is rm=(lnR0)/T. In this equation, T is the mean generation time. If there is no dataset to calculate the rm value and other life-table parameters, it is better to delete it and proceed with the discussion on survival and mortality rates.

The result of the statistical analysis in Table 1 needs to be explained. The data in Fig 7 are not statistically analyzed. Tabel 2 has the same problem, presumably the authors applied the two-way ANOVA and post-hoc multiple comparisons, but there is no explanation of this point in the Materials and Methods section as well as the caption of Table 2.

The formula for "Relative Resistance" in the last part of Materials and Methods is not very clear visually, why not replace it with a variable?

In some graphs, the labels on the X-axis overlap with the bars; please move the X-axis to an appropriate position.

Author Response

We thank the reviewers for their comments and suggestions, which improve its accessibility to general readers. The manuscript has been revised based on the suggestions provided by the reviewers. All the comments and suggestions provided by the reviewers have been included in the revised manuscript.  All the changes have been marked in the revised manuscript.

Reviewer #1 round 2

Comment: In Fig. 1a and b, it is inappropriate to connect the lines for data between the diet types. I do not understand the reason why they are separated at Day 5 and 7. Why don't you show the lx and mx curves for each diet type or sucrose content during the treatment period, and put the λ, rm, and R0 in a table? Perhaps the authors do not understand how to calculate the rm value. The survival rate per day (lx) and the number of litters (mx) must be counted until all individuals die, and the final result is rm=(lnR0)/T. In this equation, T is the mean generation time. If there is no dataset to calculate the rm value and other life-table parameters, it is better to delete it and proceed with the discussion on survival and mortality rates.

Response: Thank you for shedding light on the life-table parameters. Unfortunately, we do not have daily count data of the aphid till all individuals die. We collected the data at a maximum period of 7 days because the liquid synthetic diet is prone to fungal contamination after prolonged period. Also, we wanted to minimize induced death either due to diet contamination when aphids probe the sterile diet or frequent transfer of the aphids.  However, we have based our discussion on the survivability of banana aphids at day 7 post-feeding on various diets supplemented with different concentrations of sucrose. The Figure 1 has been revised accordingly.

Comment: The result of the statistical analysis in Table 1 needs to be explained.

Response: The result have been explained. We also like to mention that to ensure uniformity, we have used dunnett’s test in addition to the ANOVA for table 1 and 2 to compare the treatments with their respective controls. As a result, there are few changes to the levels of significance, and this have been incorporated.

Comment: The data in Fig 7 are not statistically analyzed.

Response: The result has been statistically analyzed and showed in revised Figure 7.

Comment: Table 2 has the same problem, presumably the authors applied the two-way ANOVA and post-hoc multiple comparisons, but there is no explanation of this point in the Materials and Methods section as well as the caption of Table 2.

Response:  The explanation has been included in the and materials and methods.

Comment: The formula for "Relative Resistance" in the last part of Materials and Methods is not very clear visually, why not replace it with a variable?

Response:  It has been revised.

Comment: In some graphs, the labels on the X-axis overlap with the bars; please move the X-axis to an appropriate position.

Response:  It has been corrected.

Round 3

Reviewer 1 Report

In Tables, please specify the initial sample size (n).

Fig 1: It is not desirable to make line graphs with equal intervals because the intervals of the X-label values are different.  Please change the type of the graph.

Fig. 2: I don't understand why the authors don't modify such an incomprehensible figure. I would like the editors to point out such points that need to be corrected.

Author Response

Comment 1. In Tables, please specify the initial sample size (n).

Response: It is mentioned in the revised manuscript.

Comment 2.  Fig 1: It is not desirable to make line graphs with equal intervals because the intervals of the X-label values are different.  Please change the type of the graph.

Response: The graph type has been changed to a bar graph.

Comment 3. Fig. 2: I don't understand why the authors don't modify such an incomprehensible figure. I would like the editors to point out such points that need to be corrected.

Response: The graph has been changed to a bar graph for better clarity.